

# Population-specific equations of age-related maximum handgrip force: a comprehensive review

Danuta Roman-Liu,  Joanna Kamińska and  Tomasz Macjej Tokarski

Ergonomics, Central Institute for Labour Protection–National Research Institute, Warsaw, Poland

## ABSTRACT

**Background**. The measurement of handgrip force responses is important in many aspects, for example: to complement neurological assessments, to investigate the contribution of muscle mass in predicting functional outcomes, in setting realistic treatment goals, evaluating rehabilitation strategies. Normative data about handgrip force can assist the therapist in interpreting a patient's results compared with healthy individuals of the same age and gender and can serve as key decision criteria. In this context, establishing normative values of handgrip strength is crucial. Hence, the aim of the this study is to develop a tool that could be used both in rehabilitation and in the prevention of work-related musculoskeletal disorders. This tool takes the form of population-specific predictive equations, which express maximum handgrip force as a function of age.

**Methodology**. In order to collect data from studies measuring maximum handgrip force, three databases were searched. The search yielded 5,058 articles. Upon the removal of duplicates, the screening of abstracts and the full-text review of potentially relevant articles, 143 publications which focussed on experimental studies on various age groups were considered as fulfilling the eligibility criteria. A comprehensive literature review produced 1,276 mean values of maximum handgrip force.

**Results**. A meta-analysis resulted in gender- and world region-specific (general population, USA, Europe and Asia) equations expressing maximum force as a function of age. The equations showed quantitative differences and trends in maximum handgrip force among age, gender and national groups. They also showed that values of maximum handgrip force are about 40% higher for males than for females and that age-induced decrease in force differs between males and females, with a proved 35% difference between the ages of 35 and 75. The difference was lowest for the 60–64 year olds and highest for the 18–25 year-olds. The equations also showed that differences due to region are smaller than those due to age or gender.

**Conclusions**. The equations that were developed for this study can be beneficial in setting population-specific thresholds for rehabilitation programmes and workstation exposure. They can also contribute to the modification of commonly used methods for assessing musculoskeletal load and work-related risk of developing musculoskeletal disorders by scaling their limit values.

Corresponding author
Danuta Roman-Liu, daliu@ciop.pl

# INTRODUCTION

Handgrip strength is a simple, reliable, and inexpensive test that can be used as a standalone evaluation in a variety of settings, including sports, medicine, geriatrics, rehabilitation and ergonomics. Handgrip force is a widely used indicator because it can be easily obtained and can be treated as a reliable surrogate for whole body strength (*Toselli et al., 2020*; *Voorbij & Steenbekkers, 2001*; *Kim et al., 2021*). The measurement of maximal handgrip strength is an essential component of tracing one's performance through physical training, injury rehabilitation and other therapeutic processes (*Tveter et al., 2014*). Values of handgrip strength are crucial in determining the extent of injury either in musculoskeletal or neurological structures. Low muscle strength predicts limited mobility, propensity to falls, and activity restriction (*Bae et al., 2019*; *Kim, 2021*; *Oksuzyan et al., 2017*; *Alkahtani et al., 2020*). Handgrip force can be a part of exercise sets prescribed for persons with Parkinson's disease, where the level of those exercises is important (*Gobert et al., 2021*). The measurement of reactive grip force responses may be used to complement neurological assessments or to investigate the contribution of muscle mass in predicting functional outcomes (*Di Monaco et al., 2014*). Handgrip force can be also a measure of differences in physical performance and health status due to environmental conditions that people live (*Sampaio et al., 2012*) or association of muscle strength represented by handgrip with advanced glycation end products (*Dalal et al., 2009*). Crucial topic that emphasizes the need of handgrip strength evaluations is ergonomics. The principal contributing risk factor for the development of musculoskeletal disorders (MSDs) is work-related musculoskeletal load that depends on a combination of factors, among which exerted force plays an important role (*Roman-Liu, 2013*). This means that handgrip force measurements can assist the ergonomist in assessment of musculoskeletal load and the therapist in interpreting a patient's results compared with healthy individuals of the same age and gender. They can also be used in setting realistic treatment goals, evaluating rehabilitation strategies and can serve as key decision criteria. The measurement of grip strength is included not only in the assessment of individuals who experience impaired performance of daily tasks but also has great significance for prevention purposes.

Progressive decline in strength capability in older individuals increases relative load. This means that certain levels of force exertion can be less hazardous for younger employees than for older ones. As such, part and parcel of the prevention measures that would reduce risk is the adjustment of exerted force limits depending on age. This requires quantifying age-related changes in strength capabilities. Given that strength capabilities, including age-related strength loss, are typically assessed by the maximum grip strength value (*Leong et al., 2015*; *Schettino et al., 2014*), data on the relationship between age and maximum strength values are very needed. In this context, establishing normative values of handgrip strength is crucial.

The literature emphasises the scarcity of objective, age-related standards of grip strength, while highlighting that a lot of the research has focussed on handgrip strength measurements. For the most part, norm tables are presented with data stratified by gender and age group (*Alrashdan, Ghaleb & Almobarek, 2021*; *Schlüssel et al., 2008*; *Voorbij*

*& Steenbekkers, 2001*; *Wang et al., 2019*; *Yoo, Choi & Ha, 2017*; *Yu et al., 2017*). Standard tables include a discrete set of data points, where typically one value represents a period of ten years. Predictive equations that provide more specific values have greater explanatory power.

Equations that present handgrip force as dependent on age and gender can also take into consideration other individual characteristics. *Lunde, Brewer & Garcia (1972)*, for example, derived equations to predict both dominant and non-dominant handgrip strength using body height and weight. Similarly, *Crosby, Wehbé & Mawr (1994)* proposed a mathematical relationship between maximum grip strength and anthropometric measurements (height and weight of the body). *Hanten et al. (1999)* also found grip strength to be proportional to the height and weight of one's body, as well as to their age and gender. More recently, *Tveter et al. (2014)* proposed equations for two age groups: less than 50 years of age and above 50 for the left and right hand.

The above-mentioned equations form the summation of the components which are products of the coefficients and variables relating to body height, weight, sex and age. They were created on the basis of research on a given group of subjects. It can be assumed that the consolidation of results of existing studies conducted on many different groups might be more representative. Some studies have shown that differences in age-related handgrip force are population-specific and dependent on one's country or world region (*Dodds et al., 2016*; *Alrashdan, Ghaleb & Almobarek, 2021*; *Rostamzadeh, Saremi & Bradtmiller, 2020*). This suggests that studies on a limited group of participants that belong to any one ethnicity or country might yield results specific for this world region only and that this aspect should be analysed.

Some attempts have been made to derive empirical consolidated data associated with specific age groups. *Bohannon et al. (2006)* established a consolidated table of norms assigning handgrip force values to gender and age (measured in decades), without, however, taking into account the population specificity aspect. In this light, it seems reasonable to conduct a theoretical study that would single out predictive equations for handgrip force on the basis of the available results from existing experimental studies. As the handgrip force capabilities of different age groups have already been studied, the results of those studies can be used to explore this issue further, increasing the likelihood of obtaining equations that could be used in the treatment and prevention of MSDs. Review studies that compile the research on handgrip force measurements have been published. Eight publications on both patients and healthy people were included in the most current umbrella review (*Soysal et al., 2021*). This article adds significantly to our understanding of handgrip strength as a predictor of overall health, disability, and cardiovascular mortality. Nevertheless, neither this study nor the articles referenced in this publication have not quantified how aging impacts handgrip strength.

The aim of this study is to develop a tool, in the form of population-specific predictive equations, that expresses maximum handgrip strength as a function of age, which could be used in the rehabilitation and prevention of work-related MSDs. Based on the meta-analysis of existing data from experimental studies, equations were developed specific to the general population of males and females and to world regions. These equations will provide pointers

as to the degree of influence of age, gender and world regions on maximum handgrip force, and determine which of these factors has the greatest impact on strength decline as a direct result of the ageing process. This indicates that the developed equations meet the goal of protecting the health of the elderly population, as their use can be beneficial in setting population-specific thresholds for workstation exposure and rehabilitation programs.

## METHODOLOGY

### Literature review

A comprehensive literature review was conducted in step with the review protocol which involves three stages: implementing a systematic review protocol, employing the strategy designed by the PRISMA checklist (*Page et al., 2021*), and summarizing the extracted findings from the eligible studies. This included the formulation of the research question, the selection of bibliographic databases and search strings to be used, and the inclusion and exclusion criteria to be applied, both for searching the databases and for analysing the information retrieved.

The following research questions were was formulated: how can the available literature on maximum force capabilities be characterised and used to develop equations expressing maximum handgrip force as a function of age with and without consideration of nationality; what is the impact of gender and world regions on strength decline with age?

ScienceDirect, all the databases of ProQuest and PubMed were selected for review in terms of titles and abstracts, using the search string "handgrip". The search included full-text articles published in English. No additional restrictions were applied in ScienceDirect. In PubMed, search was limited to case reports, foundational articles, clinical studies, journal articles and human studies. The search in ProQuest included research journals, papers, case studies and peer-reviewed full-text articles. Additionally, the reference lists of analysed articles, including reviews, were scanned to identify non-indexed articles that might meet the eligibility criteria. Two authors independently performed title and abstract screening. Disagreements among reviewers were resolved through discussion or by a third reviewer. Screening full-texts for eligibility was done in a similar manner.

The next step consisted in a preliminary analysis of the retrieved articles in order to select those fulfilling the following inclusion criteria: (a) the study presented results of handgrip force measurements for male and/or female populations; (b) the age of the participants could be identified from means or ranges of values; (c) healthy participants were aged between 18 and 80 years old. Excluded studies: (a) presented results for males and females as a mixed group, or for transgender people; (b) calculated the mean value of force for an age range greater than 15 years or gave a standard deviation of mean age higher than 6 years; (c) did not state the force units; (d) presented duplicated results already included in the analysis. Articles that presented groups affected by disease and healthy controls were accepted with reference to the healthy groups only.

### Approximation of numerical data

Eligible data were extracted in Microsoft Excel and then exported from Microsoft Excel into STATISTICA 12. All data collected were divided into groups according to the participants'

age and gender. Each group spanned five years, apart from the youngest group that covered ages from 18 to 24. Qualification to an age group was based on an age range or on the mean value of age. For each age group, the mean value for grip strength, sample size and country of population under study was assigned. Handgrip force values in pounds or kilograms were converted to Newtons.

Sets of structured data, separate for males and females, which showed the values of maximum handgrip force and respective age group were averaged with the weighted average protocol. Averaged values within age groups were approximated with third-degree polynomial regressions. In this way, gender-specific equations were obtained for the general population. In order to obtain region-specific equations from gender groups, subgroups of citizens of the USA, European countries and Asian countries were extracted. Data in each of these subgroups formed the basis for six additional region-specific equations.

In order to evaluate the applicability of the equations, values of handgrip force obtained from studies of consolidated data of handgrip force, not included in the analysis, were used.

### Risk of bias assessment

The assessment of publication bias was conducted by the determination of statistical significance accomplished by employing the Begg and Mazumdar's test (*Begg & Mazumdar, 1994*). Test reports the rank correlation (Kendall's tau) between the standardized effect size and the variances of these effects, which tests whether the standardized effect size and the variance of the effect size are significantly associated. In the absence of publication bias the test statistic is assumed to be close to zero. In the presence of publication bias, a test statistic is significantly greater than zero. Kendall's tau, Z statistics, and p level were determined for each of the age and gender groups and presented on a forest plot that demonstrated the mean and standard deviation of each study variant that was under analysis.

## RESULTS

### Literature review

The search produced 5,058 articles (1,535 from ScienceDirect, 2,041 from ProQuest and 1,482 from PubMed) (Fig. 1). The removal of duplicates and the screening of titles and abstracts of potentially relevant articles resulted in 453 publications eligible for full-text review.

Following a final comprehensive analysis of the articles, the numerical values of maximum handgrip force from 143 publications were derived. Those publications presented data divided into male and female populations. Tables 1 to 4 summarise the main characteristics of the studies which met all criteria. Tables 1 and 2 summarise studies conducted on both male and female groups, whereas Tables 3 and 4 include studies conducted only on males or only on females.

Studies were conducted on specific populations with a number of samples in each age group ranging from a few up to a few hundred. This provided a large amount of data available for further analysis. Among those studies, the highest number of publications was affiliated to the United States (*Bemben & Langdon, 2002*; *Blackwell, Kornatz & Heath, 1999*;

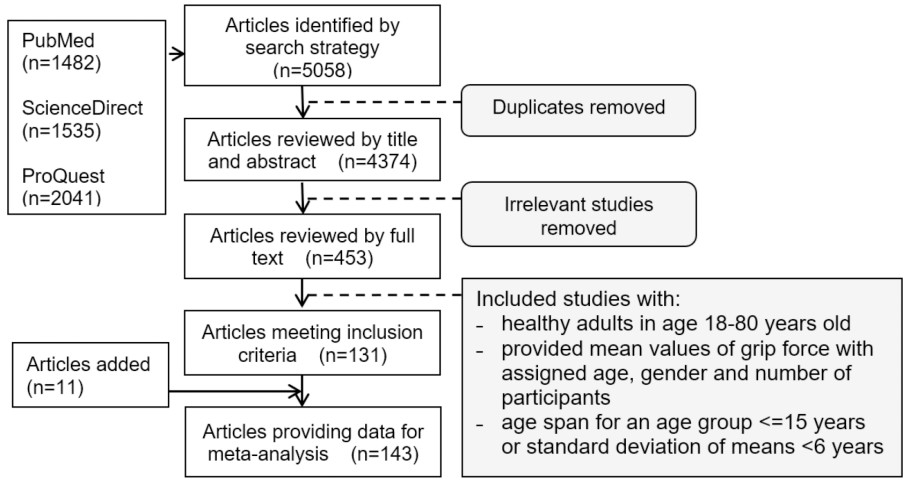

**Figure 1** Flow chart illustrating the results of the comprehensive literature review.

*Dale et al., 2014*; *Dunn-Lewis et al., 2011*; *Forrest, Zmuda & Cauley, 2005*; *Forrest, Zmuda & Cauley, 2007*; *Garg et al., 2002*; *Hanten et al., 1999*; *Kallman, Plato & Tobin, 1990*; *Kaiser, 2000*; *Mathiowetz et al., 1985*; *Mehta & Cavuoto, 2015*; *Meyers, 2006*; *Nicolay & Walker, 2005*; *Sella, 2001*; *Stewart et al., 2009*; *Taaffe & Marcus, 2004*; *Wang et al., 2019*; *Weinstock-Zlotnick, Bear-Lehman & Yu, 2011*) and Japan (*Abe, Thiebaud & Loenneke, 2016*; *Hioka et al., 2021*; *Kubota, Demura & Kawabata, 2012*; *Kwon et al., 2017*; *Murata et al., 2010*; *Osei-Hyiaman et al., 1999*; *Rantanen et al., 1998*; *Seino et al., 2014*; *Shimizu et al., 2021*; *Shimizu et al., 2018*; *Sugiura et al., 2013*; *Sun et al., 2010*; *Yamagata & Sako, 2020*; *Yoshimura et al., 2011*). A lower number was attributed to Brazil (*Gallo et al., 2019*; *Heberle et al., 2021*; *Vilaça et al., 2014*; *Aguiar et al., 2018*; *de Branco et al., 2020*; *Dias et al., 2012*; *Lopes et al., 2018*; *Prata & Scheicher, 2015*; *Schettino et al., 2014*; *Silva et al., 2013*), India (*Chatterjee & Chowdhuri, 1991*; *Choudhary et al., 2016*; *Gunasekaran et al., 2016*; *Kaur, 2009*; *Khanna & Koley, 2020*; *Koley & Kaur, 2011*; *Kulkarni et al., 2014*; *Parvatikar & Mukkannavar, 2009*; *Saha, 2013*), the United Kingdom (*Baylis et al., 2014*; *Faber, Hansen & Christensen, 2006*; *Gedmantaite et al., 2020*; *Oksuzyan et al., 2017*; *Skelton et al., 1994*; *Stevens et al., 2012*; *Syddall et al., 2017*) and China (*Auyeung et al., 2014*; *Shen et al., 2016*; *Song et al., 2020*; *Wang et al., 2021*; *Wu et al., 2012*; *Zeng et al., 2015*) respectively. Five publications concerned Iran (*Abiri, Dehghani & Vafa, 2020*; *Kheyruri et al., 2021*; *Mohammadian et al., 2014*; *Moslehi et al., 2013*; *Rostamzadeh, Saremi & Bradtmiller, 2020*), Italy (*Bergamin et al., 2015*; *Campa et al., 2021*; *Montalcini et al., 2013*; *Sarti et al., 2013*; *Zaccagni et al., 2020*), Korea (*Kim et al., 2021*; *Lee & Lee, 2013*; *Shim et al., 2013*; *Yoo, Choi & Ha, 2017*; *Sin et al., 2009*) and the Netherlands (*Leenders et al., 2013*; *de Vreede et al., 2005*; *Ling et al., 2012*; *Sialino et al., 2019*; *Voorbij & Steenbekkers, 2001*) respectively. Next came Malaysia (*Lam et al., 2016*; *Ing et al., 2018*; *Justine, Nawawi & Ishak, 2020*; *Moy, Chang & Kee, 2011*), Poland (*Fiutko, 1987*; *Kociuba et al., 2019*; *Kopiczko, Gryko & Łopuszańska-Dawid, 2018*; *Roman-Liu, Tomasz Tokarski & Mazur-Różycka, 2021*) and Taiwan (*Kao et al., 2016*; *Lin et al., 2014*; *Su et al., 1994*; *Wu et al., 2009*) with four publications each, while Belgium

**Table 1 Characteristics of studies that presented results for male (M) and female (F) populations: the number of participants (n) in each group and their age (range).**

| References | Age | n F | n M | References | Age | n F | n M | References | Age | n F | n M |
|---|---|---|---|---|---|---|---|---|---|---|---|
| Abaraogu et al. (2017) | 18–25 | 237 | 280 | Mendes et al. (2017) | 65–75 | 423 | 354 | | 21–30 | 85 | 80 |
| | 18–29 | 40 | 42 | | 20–24 | 44 | 55 | | 31–40 | 108 | 78 |
| | 30–39 | 40 | 41 | | 25–29 | 42 | 43 | Tsang (2005) | 41–50 | 79 | 38 |
| Abe, Thiebaud & Loenneke (2016) | 40–49 | 43 | 43 | | 30–34 | 42 | 37 | | 51–60 | 33 | 23 |
| | 50–59 | 47 | 47 | | 35–39 | 42 | 40 | | 61–70 | 14 | 4 |
| | 60–69 | 47 | 46 | | 40–44 | 42 | 39 | Veen et al. (2021) | 65–70 | 122 | 71 |
| | 70–79 | 48 | 47 | Mohammadian et al. (2014) | 45–49 | 43 | 43 | | 20–30 | 68 | 55 |
| | 20–29 | 262 | 280 | | 50–54 | 44 | 51 | | 50–54 | 35 | 35 |
| Adedoyin et al. (2009) | 30–39 | 47 | 79 | | 55–59 | 39 | 41 | | 55–59 | 50 | 46 |
| | 40–49 | 12 | 28 | | 60–64 | 40 | 46 | Voorbij & Steenbekkers (2001) | 60–64 | 53 | 44 |
| | 50–59 | 10 | 15 | | 65–69 | 37 | 39 | | 65–69 | 51 | 50 |
| | 60–69 | 5 | 7 | | 70–74 | 34 | 46 | | 70–74 | 62 | 59 |
| | 20–29 | 30 | 30 | Montalcini et al. (2013) | 19–25 | 178 | 157 | | 75–79 | 38 | 36 |
| Alrashdan, Ghaleb & Almobarek (2021) | 30–39 | 30 | 30 | Nicolay & Walker (2005) | 18–33 | 33 | 17 | | 20–24 | 443 | 471 |
| | 40–49 | 30 | 30 | | 55–59 | 1364 | 1315 | | 25–29 | 382 | 380 |
| | 50–59 | 30 | 30 | | 60–64 | 1265 | 1156 | | 30–34 | 413 | 418 |
| | 60–69 | 26 | 30 | Oksuzyan et al. (2017) | 65–69 | 1053 | 990 | | 35–39 | 396 | 363 |
| | 65–69 | 537 | 567 | | 70–74 | 1333 | 1287 | Wang et al. (2019) | 40–44 | 421 | 387 |
| Auyeung et al. (2014) | 70–74 | 501 | 560 | | 75–79 | 867 | 776 | | 45–49 | 385 | 357 |
| | 75–79 | 312 | 295 | | 25–34 | 10 | 13 | | 50–54 | 416 | 365 |
| | 20–24 | 31 | 80 | Peolsson, Hedlund & Oberg (2001) | 35–44 | 15 | 15 | | 55–59 | 352 | 347 |
| | 25–29 | 51 | 102 | | 45–54 | 14 | 12 | | 60–64 | 413 | 407 |
| | 30–34 | 30 | 80 | | 55–64 | 11 | 11 | | 65–69 | 295 | 280 |
| Bernardes et al. (2020) | 35–39 | 22 | 73 | Parvatikar & Mukkannavar (2009) | 18–25 | 50 | 50 | | 70–74 | 282 | 226 |
| | 40–44 | | 54 | Pieterse, Manandhar & Ismail (2002) | 50–59 | 189 | 189 | | 75–79 | 153 | 161 |
| | 45–49 | 34 | 60 | | 60–69 | 156 | 153 | Weinstock-Zlotnick, Bear-Lehman & Yu (2011) | 18–24 | 80 | 66 |
| | 50–54 | | | Ramlagan, Peltzer & Phaswanan-Mafuya (2014) | 50–59 | 949 | 746 | | | | |
| | 20–29 | 5 | 5 | | 60–69 | 690 | 543 | Wiraguna & Setiati (2018) | 60–74 | 41 | 39 |
| | 30–39 | 5 | 5 | | 70–79 | 370 | 291 | | | | |
| Bowden & McNulty (2013) | 42–49 | 5 | 5 | | 60–64 | 24 | 1 | | 20–24 | 31 | 29 |
| | 50–59 | 5 | 5 | | 65–69 | 109 | 9 | | 25–29 | 30 | 30 |
| | 60–69 | 5 | 5 | Río et al. (2020) | 70–74 | 316 | 35 | | 30–34 | 30 | 28 |
| | 71–78 | 5 | 5 | | 75–79 | 488 | 76 | | 35–39 | 42 | 41 |
| Bucci et al. (2013) | 18–30 | 78 | 74 | Rodrigues-Barbosa et al. (2011) | 60–69 | 824 | 623 | | 40–44 | 39 | 37 |
| | 69–81 | 132 | 128 | | 70–79 | 641 | 426 | | 45–49 | 40 | 31 |
| Charalabos et al. (2013) | 18–24 | 180 | 180 | | | | | | | | |

| References | Age | F | M | References | Age | F | M | References | Age | F | M |
|---|---|---|---|---|---|---|---|---|---|---|---|
| *Chilima & Ismail (2000)* | 55–59 | 51 | 12 | *Roman-Liu, Tomasz Tokarski & Mazur-Rózycka (2021)* | 20–30 | 60 | 60 | *Werle et al. (2009)* | 50–54 | 34 | 40 |
| | 60–69 | 111 | 40 | | 55–67 | 60 | 60 | | 55–59 | 28 | 30 |
| *De Smet, Tirez & Stappaerts (1998)* | 19–30 | 20 | 20 | | 20–24 | 198 | 188 | | 60–64 | 30 | 33 |
| *Desrosiers et al. (1995)* | 60–69 | 59 | 61 | | 25–29 | 175 | 173 | | 65–69 | 34 | 46 |
| | 70–79 | 60 | 60 | | 30–34 | 180 | 186 | | 70–74 | 27 | 33 |
| *Faber, Hansen & Christensen (2006)* | 18–29 | 48 | 52 | | 35–39 | 178 | 173 | | 75–79 | 26 | 28 |
| | 30–44 | 86 | 81 | | 40–44 | 151 | 191 | | 20–24 | 21 | 41 |
| *Gunasekaran et al. (2016)* | 60–64 | 185 | 330 | *Rostamzadeh, Saremi & Bradtmiller (2020)* | 45–49 | 180 | 180 | | 25–29 | 21 | 20 |
| | 65–69 | 34 | 95 | | 50–54 | 167 | 175 | | 30–34 | 14 | 19 |
| | 20–29 | 75 | 66 | | 55–59 | 192 | 170 | | 35–39 | 17 | 19 |
| | 30–39 | 70 | 52 | | 60–64 | 156 | 167 | | 40–44 | 27 | 19 |
| *Günther et al. (2008)* | 40–49 | 63 | 62 | | 65–69 | 178 | 183 | *Wu et al. (2009)* | 45–49 | 23 | 19 |
| | 50–59 | 61 | 62 | | 70–74 | 169 | 178 | | 50–54 | 25 | 22 |
| | 60–69 | 64 | 65 | | 75–79 | 191 | 203 | | 55–59 | 16 | 15 |
| | 70–79 | 42 | 47 | | 20–29 | 431 | 295 | | 60–64 | 20 | 13 |
| | 20–24 | 80 | 74 | | 30–39 | 397 | 244 | | 65–69 | 14 | 15 |
| | 25–29 | 90 | 104 | *Schlüssel et al. (2008)* | 40–49 | 403 | 220 | | 70–74 | 17 | 18 |
| | 30–34 | 88 | 62 | | 50–59 | 327 | 166 | | 50–54 | 329 | 211 |
| | 35–39 | 76 | 60 | | 60–69 | 198 | 121 | | 55–59 | 294 | 169 |
| *Hanten et al. (1999)* | 40–44 | 71 | 55 | | 65–69 | 588 | 481 | *Wu et al. (2012)* | 60–64 | 239 | 176 |
| | 45–49 | 72 | 52 | *Seino et al. (2014)* | 70–74 | 847 | 727 | | 65–69 | 160 | 143 |
| | 50–54 | 56 | 52 | | 75–79 | 658 | 615 | | 70–74 | 108 | 122 |
| | 55–59 | 59 | 51 | | 20–29 | 63 | 42 | *Yeung et al. (2018)* | 18-30 | 96 | 85 |
| | 60–64 | 49 | 51 | | 30–39 | 79 | 86 | | 69-81 | 159 | 161 |
| | 20–29 | 32 | 27 | | 40–49 | 102 | 150 | | 19–24 | 199 | 186 |
| | 30–39 | 31 | 32 | *Sella (2001)* | 50–59 | 93 | 129 | | 25–29 | 140 | 143 |
| *Hogrel (2015)* | 40–49 | 32 | 26 | | 60–69 | 21 | 66 | | 30–34 | 200 | 132 |
| | 50–59 | 29 | 11 | | 70–79 | 15 | | | 35–39 | 216 | 166 |
| | 60–69 | 21 | 11 | | 20–29 | 36 | 33 | | 40–44 | 248 | 211 |
| | 70–79 | 17 | 5 | | 30–39 | 32 | 14 | *Yoo, Choi & Ha (2017)* | 45–49 | 264 | 156 |
| *Ing et al. (2018)* | 60–69 | 90 | 47 | *Shim et al. (2013)* | 40–49 | 55 | 28 | | 50–54 | 273 | 189 |
| | 70–79 | 42 | 34 | | 50–59 | 40 | 27 | | 55–59 | 275 | 234 |
| | 65–69 | 134 | 99 | | 60–69 | 7 | 13 | | 60–64 | 221 | 165 |
| *Jais et al. (2018)* | 70–74 | 134 | 99 | | 70–79 | 5 | 8 | | 65–69 | 184 | 165 |
| | 75–79 | 134 | 99 | *Sialino et al. (2019)* | 55–65 | 1756 | 1713 | | 70–74 | 145 | 102 |
| *Khanna & Koley (2020)* | 18–25 | 128 | 100 | | 60–69 | 138 | 64 | | 75–79 | 107 | 92 |
| | 60–64 | 74 | 50 | *Silva et al. (2013)* | 70–79 | 93 | 42 | | 40–49 | 47 | 16 |
| *Lam et al. (2016)* | 65–69 | 46 | 30 | | 65–69 | 10 | 10 | *Yoshimura et al. (2011)* | 50–59 | 96 | 50 |
| | 70–74 | 49 | 30 | *Skelton et al. (1994)* | 70–74 | 10 | 10 | | 60–69 | 158 | 69 |
| | 60–64 | 74 | 50 | | 75–79 | 10 | 10 | | 70–79 | 300 | 154 |

**Table 1** (*continued*)

| References | Age | n F | n M | References | Age | n F | n M | References | Age | n F | n M |
|---|---|---|---|---|---|---|---|---|---|---|---|
| Lopes et al. (2018) | 20–29 | 10 | 10 | Su et al. (1994) | 20–29 | 16 | 16 | Yu et al. (2017) | 20–29 | 39 | 15 |
| | 30–39 | 10 | 10 | | 30–39 | 16 | 16 | | 30–39 | 134 | 32 |
| | 40–49 | 10 | 10 | | 40–49 | 16 | 16 | | 40–49 | 1334 | 54 |
| | 50–59 | 10 | 10 | | 50–59 | 16 | 16 | | 50–59 | 1188 | 182 |
| Mathiowetz et al. (1985) | 20–24 | 26 | 29 | | 60–69 | 16 | 16 | | 60–69 | 556 | 230 |
| | 25–29 | 27 | 27 | Tveter et al. (2014) | 18–29 | 25 | 23 | | 70–79 | 300 | 135 |
| | 30–34 | 26 | 27 | | 30–39 | 26 | 24 | Zaccagni et al. (2020) | 18–30 | 188 | 356 |
| | 35–39 | 25 | 25 | | 40–49 | 28 | 26 | | | | |
| | 40–44 | 31 | 26 | | 50–59 | 27 | 32 | Zeng et al. (2015) | 45–54 | 43 | 17 |
| | 45–49 | 25 | 28 | | 60–69 | 29 | 25 | | 55-64 | 115 | 42 |
| | 50–54 | 25 | 25 | | 70–79 | 37 | 30 | | 65–74 | 129 | 130 |
| | 55–59 | 25 | 21 | | | | | | | | |
| | 60–64 | 25 | 24 | | | | | | | | |
| | 65–69 | 28 | 27 | | | | | | | | |
| | 70–74 | 29 | 26 | | | | | | | | |

**Table 2** Characteristics of studies that presented results for male (M) and female (F) populations: the number of participants (n) in each group and their age (mean).

| References | Age | nF | Age | nM | References | Age | nF | Age | nM |
|---|---|---|---|---|---|---|---|---|---|
| Baylis et al. (2014) | 67.2 | 95 | 67.1 | 158 | Moy, Chang & Kee (2011) | 68.6 | 213 | 67.3 | 221 |
| | 76.2 | 95 | 76.5 | 157 | Murphy et al. (2021) | 72.0 | 55 | 72.0 | 52 |
| De Dobbeleer et al. (2019) | 52.0 | 427 | 56.0 | 535 | Račić, Pavlović & Ivković (2019) | 72.7 | 159 | 73.8 | 141 |
| Dunn-Lewis et al. (2011) | 52.0 | 15 | 56.0 | 16 | Ridgway et al. (2011) | 25.5 | 401 | 25,7 | 382 |
| Gedmantaite et al. (2020) | 63.6 | 34187 | 64.0 | 33815 | Shen et al. (2016) | 18.9 | 397 | 19.1 | 211 |
| Justine, Nawawi & Ishak (2020) | 67.0 | 398 | 67.0 | 178 | Bergamin et al. (2015) | 73.0 | 230 | 75.5 | 151 |
| Kaj et al. (2015) | 21.0 | 4560 | 21.0 | 3540 | Song et al. (2020) | 67.2 | 524 | 69.0 | 285 |
| | 35.0 | 309 | 35.0 | 123 | Stevens et al. (2012) | 68.1 | 280 | 67.8 | 349 |
| Kao et al. (2016) | 72.9 | 107 | 78.3 | 98 | Sugiura et al. (2013) | 72.2 | 227 | 72.3 | 109 |
| Kim et al. (2021) | 60,8 | 5098 | 60,7 | 4131 | Sun et al. (2010) | 73.1 | 376 | 72.9 | 195 |
| Kociuba et al. (2019) | 21.0 | 76 | 21.6 | 173 | Syddall et al. (2017) | 67.2 | 120 | 67.1 | 172 |
| Kulkarni et al. (2014) | 21.0 | 416 | 21.0 | 968 | | 76.2 | 120 | 76.5 | 172 |
| Lin et al. (2014) | 72.8 | 221 | 74.7 | 251 | Wang et al. (2021) | 67.7 | 194 | 69.6 | 107 |
| Ling et al. (2012) | 64.0 | 339 | 64.0 | 333 | Woo et al. (2014) | 72.0 | 1030 | 72.8 | 976 |
| Mgbemena et al. (2019) | 20.5 | 200 | 21.7 | 200 | Wu et al. (2014) | 71.7 | 154 | 74.4 | 215 |

**Table 3** Characteristics of studies that presented results for male or female populations: the number of participants (n) in each group and their age (range).

| FEMALES | | | MALES | | |
| --- | --- | --- | --- | --- | --- |
| References | Age | n | References | Age | n |
| Abiri, Dehghani & Vafa (2020) | 40–55 | 71 | | 20–29 | 25 |
| Aguiar et al. (2018) | 18–30 | 36 | Chatterjee & Chowdhuri (1991) | 30–39 | 8 |
| Bemben & Langdon (2002) | 53–65 | 40 | | 40–49 | 7 |
| Bergamin et al. (2015) | 59–66 | 25 | Ekşioğlu (2011) | 21–33 | 12 |
| Choudhary et al. (2016) | 25–40 | 90 | | 20–24 | 160 |
| Forrest, Zmuda & Cauley (2007) | 65–69 | 4019 | Fiutko (1987) | 25–29 | 187 |
| | 70–74 | 2949 | | 30–39 | 317 |
| | 75–79 | 1481 | | 40–49 | 171 |
| Garg et al. (2002) | 21–33 | 12 | Forrest, Zmuda & Cauley (2005) | 60–64 | 78 |
| | 40–45 | 100 | | 65–69 | 75 |
| | 46–50 | 100 | | 70–74 | 65 |
| | 51–55 | 100 | | 25–39 | 31 |
| Kaur (2009) | 56–60 | 100 | | 40–44 | 344 |
| | 61–65 | 100 | | 45–49 | 335 |
| | 66–70 | 100 | Forrest et al. (2012) | 50–54 | 265 |
| Kheyruri et al. (2021) | 40–55 | 83 | | 55–59 | 234 |
| Koley & Kaur (2011) | 18–25 | 201 | | 60–64 | 210 |
| Meyers (2006) | 18–32 | 25 | | 65–69 | 154 |
| Moslehi et al. (2013) | 40–55 | 69 | | 70–74 | 80 |
| | 65–69 | 17 | Green & Stannard (2010) | 19–32 | 9 |
| Murata et al. (2010) | 70–74 | 16 | | 20–29 | 55 |
| | 75–79 | 15 | | 30–39 | 139 |
| | 40–49 | 29 | Kallman, Plato & Tobin (1990) | 40–49 | 220 |
| Osei-Hyiaman et al. (1999) | 50–59 | 281 | | 50–59 | 292 |
| | 60–69 | 227 | | 60–69 | 229 |
| | 20–30 | 20 | | 70–79 | 195 |
| | 50–64 | 19 | Kopiczko, Gryko & Łopuszańska-Dawid (2018) | 20–30 | 172 |
| Schettino et al. (2014) | | | | 45–49 | 1831 |
| | 65–74 | 27 | | 50–54 | 2792 |
| | 75–86 | 19 | Rantanen et al. (1998) | 55–59 | 159 |
| von Hurst, Conlon & Foskett (2013) | 19–29 | 137 | | 60–64 | 1334 |
| | | | | 65–69 | 450 |
| Yamagata & Sako (2020) | 18–22 | 20 | Saha (2013) | 18–25 | 290 |
| | | | Shimizu et al. (2018), Shimizu et al. (2021) | 60–69 | 257 |
| | | | Sutter et al. (2019) | 20–30 | 100 |

**Table 4  Characteristics of studies that presented results for male or female populations: the number of participants (n) in each group and their age (mean).**

| FEMALES | | | | | | MALES | | |
|---|---|---|---|---|---|---|---|---|
| References | Age | n | References | Age | n | References | Age | n |
| *de Branco et al. (2020)* | 60.7 | 34 | | 23.1 | 9 | *Blackwell, Kornatz & Heath (1999)* | 25.8 | 18 |
| *de Vreede et al. (2005)* | 74.0 | 98 | *Mehta & Cavuoto (2015)* | 25.3 | 13 | | | |
| *Gallo et al. (2019)* | 68.0 | 42 | | 54.8 | 12 | *Campa et al. (2021)* | 67.4 | 33 |
| *Heberle et al. (2021)* | 71.7 | 26 | | 66.4 | 11 | *Dias et al. (2012)* | 22.5 | 40 |
| *Hioka et al. (2021)* | 77.9 | 71 | *Moratalla-cecilia et al. (2016)* | 55.6 | 67 | *Dulac et al. (2018)* | 72.8 | 21 |
| | 20.9 | 50 | *Prata & Scheicher (2015)* | 72.4 | 11 | *Kaiser (2000)* | 29.0 | 527 |
| *Kubota, Demura & Kawabata (2012)* | 72.9 | 50 | *Ribom et al. (2010)* | 68.0 | 80 | *Leenders et al. (2013)* | 70.0 | 32 |
| *Kwon et al. (2017)* | 76.8 | 79 | *Sarti et al. (2013)* | 70.9 | 92 | *Ribom et al. (2009)* | 75.6 | 2910 |
| *Lee & Lee (2013)* | 52.4 | 148 | *Stewart et al. (2009)* | 54.3 | 242 | *Tassiopoulos & Nikolaidis (2013)* | 24.5 | 31 |
| *Marques et al. (2011)* | 69.0 | 60 | *Taaffe & Marcus (2004)* | 19.5 | 40 | | | |
| | | | *Vilaça et al. (2014)* | 69.6 | 75 | | | |

(*De Smet, Tirez & Stappaerts, 1998*; *De Dobbeleer et al., 2019*; *Ridgway et al., 2011*), Sweden (*Peolsson, Hedlund & Oberg, 2001*; *Ribom et al., 2010*; *Veen et al., 2021*), Hong Kong (*Yu et al., 2017*; *Tsang, 2005*; *Woo et al., 2014*), Nigeria (*Abaraogu et al., 2017*; *Adedoyin et al., 2009*; *Mgbemena et al., 2019*) and Portugal (*Bernardes et al., 2020*; *Marques et al., 2011*; *Mendes et al., 2017*) followed with three publications on maximum handgrip force. Data for handgrip force in countries such as Canada (*Desrosiers et al., 1995*; *Dulac et al., 2018*), Spain (*Moratalla-Cecilia et al., 2016*; *Río et al., 2020*), France (*Hogrel, 2015*; *Sutter et al., 2019*), Greece (*Charalabos et al., 2013*; *Tassiopoulos & Nikolaidis, 2013*), and New Zealand (*Green & Stannard, 2010*; *von Hurst, Conlon & Foskett, 2013*) were represented in two publications per country. One publication dealt with data representative for Australia (*Bowden & McNulty, 2013*), Barbados and Cuba (*Rodrigues-Barbosa et al., 2011*), the Caribbean Island of Tobago (*Forrest et al., 2012*), Denmark (*Oksuzyan et al., 2017*), Germany (*Günther et al., 2008*), Hungary (*Kaj et al., 2015*), Indonesia (*Wiraguna & Setiati, 2018*), Ireland (*Murphy et al., 2021*), Kuwait (*Fiutko, 1987*), Malawi (*Chilima & Ismail, 2000*), Norway (*Tveter et al., 2014*), Russia (*Oksuzyan et al., 2017*), Rwanda (*Pieterse, Manandhar & Ismail, 2002*), Bosnia (*Račić, Pavlović & Ivković, 2019*), Saudi Arabia (*Alrashdan, Ghaleb & Almobarek, 2021*), Singapore (*Jais et al., 2018*), South Africa (*Ramlagan, Peltzer & Phaswanan-Mafuya, 2014*), Switzerland (*Werle et al., 2009*) and Turkey (*Ekşioğlu, 2011*). Finally, two studies cluster together a few European countries (*Bucci et al., 2013*; *Yeung et al., 2018*).

## Approximation of numerical data–mathematical relationships describing age-related fluctuations in maximum handgrip force

Study selection resulted in 1,276 mean values of maximum handgrip force measured in populations differentiated by age and gender. Taking into account the number of participants in each study, the values of maximum handgrip force from 202,361

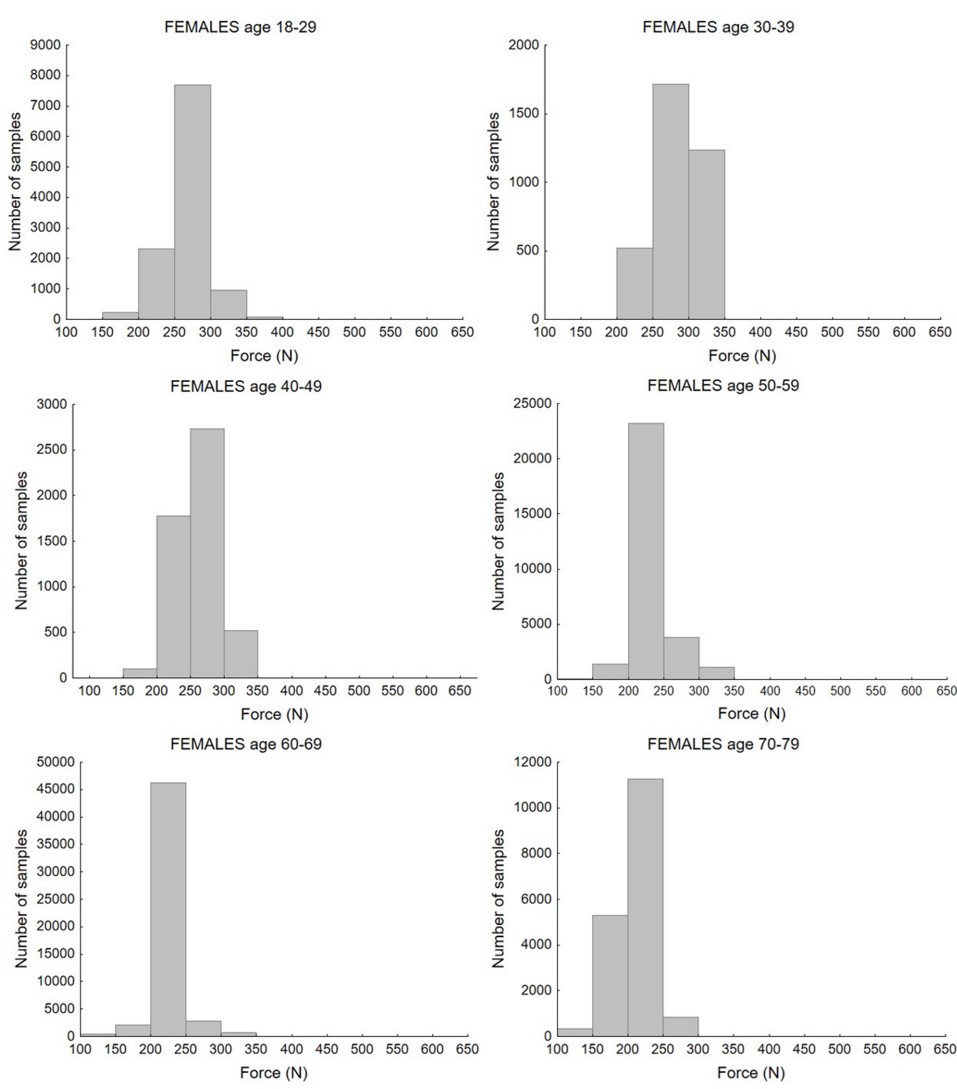

**Figure 2  Distribution of maximum force values according to age group for the female population.**

individuals were analysed. There were 666 mean values with 102,018 total values for the female population and, respectively, 610 mean values and 99,773 total values for the male population. Figures 2 and 3 present histograms of data distribution by age group and gender.

The histograms show the distribution of maximum force values by age group. Most values of strength for the female population, in age groups between 18 and 49 years old, ranged from 200 to 350 N. In the 60–79 group, the majority of cases fell below 250 N. For the male population, the highest number of samples ranged from 400 to 550 N in the 18–29 and 30–39 age groups. For the rest of the age groups, most values were obtained in the 350–400 N range, while for the oldest age group, 70–79 years old, these ranged between

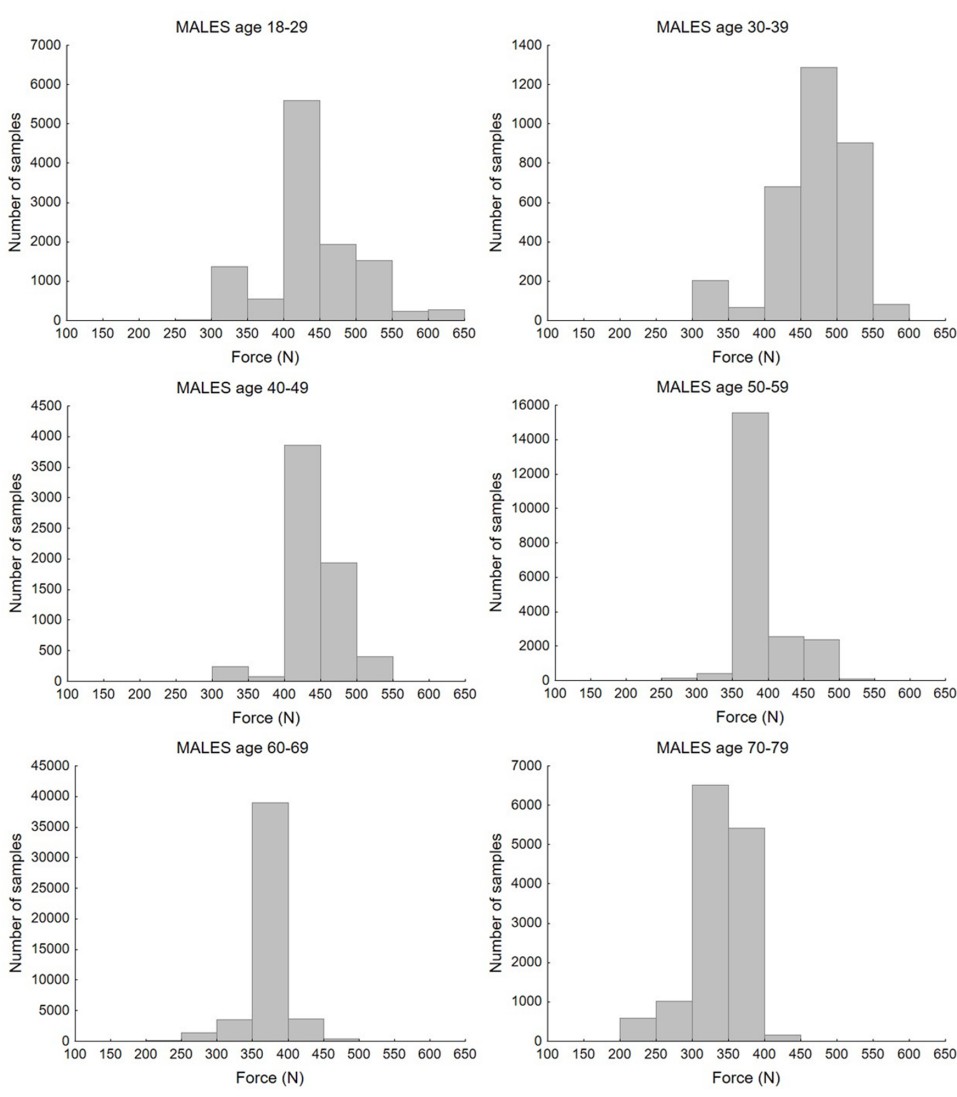

**Figure 3  Distribution of maximum force values according to age group for the male population.**

300–400 N. Bar charts that illustrate differences between age and sex groups are presented in Fig. 4.

The forest plots presenting mean values, standard deviation of the maximum force values, and weighted average for each age group of males and females are presented as supplementary data. Figures S1 to S5 present values for females and Figs. S6 to S10 for males. To assess bias associated with that group, Kendall's tau with Z statistics and p levels are employed; the results are shown in supplementary figures. In all age groups, there is a lack of significance for the female population, indicating the absence of bias. The greatest Z statistic value (1,74) is seen in the age range of 18 to 24 years old. For the male population, this age group similarly has the highest Z and tau statistics with $p < 0.05$.

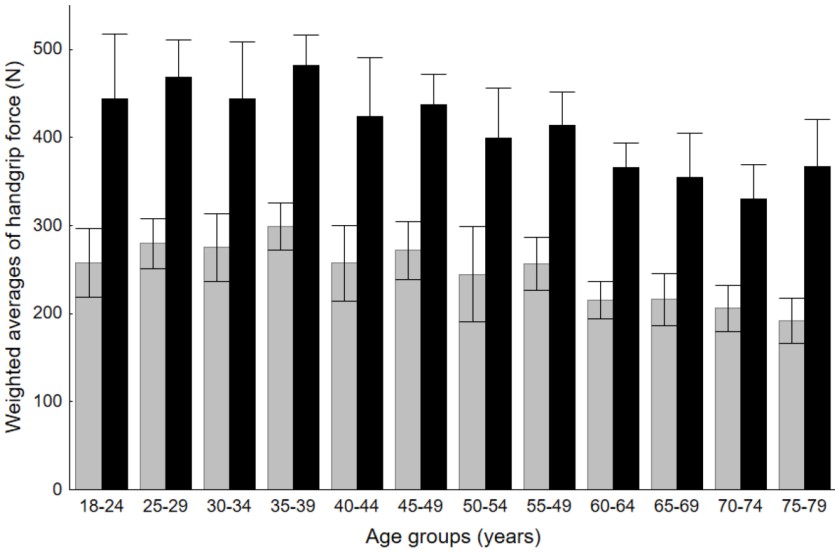

**Figure 4** Comparison of handgrip force between males and females in respective age groups.

**Table 5** Gender- and world region-specific equations presenting quantitative changes in maximum force capabilities according to age.

| World region | Females | Males |
|---|---|---|
| Overall | $= 71+12.8\cdot age-0.24\cdot age^2+0.0012\cdot age^3$ | $= 180+20\cdot age-0.415\cdot age^2+0.0023\cdot age^3$ |
| Europe | $= 22+19\cdot age-0.367\cdot Age^2+0.002\cdot age^3$ | $= 125+24\cdot age-0.45\cdot age^2+0.0023\cdot age^3$ |
| USA | $= 5+19\cdot age-0.3677\cdot age^2+0.002\cdot age^3$ | $= 143+21.73\cdot age-0.43\cdot age^2+0.0023\cdot age^3$ |
| Asia | $= 155+4.8\cdot age-0.068\cdot age^2+0.0001\cdot age^3$ | $= 250+12\cdot age-0.24\cdot age^2+0.0012\cdot age^3$ |

Predictive equations were developed for the general male and female populations and for specific world regions. World region-specific equations were developed for the United States, Europe and Asia. Table 5 presents all equations.

Figures 5 and 6 show the maximum force values for female and male populations respectively. They present the scattered values of maximum force as documented in the reviewed studies, the weighted averages obtained on the basis of the meta-analysis for specific age groups, and the outcomes of the calculations of the developed functions obtained through data approximation. The results of the equations were compared with experimental data from published studies (19 publications for the USA, 35 for Europe and 32 for Asia).

Figures 5 and 6 demonstrate high agreement between curves that illustrate values calculated by the developed models and the values corresponding to weighted averages. Correlations between those values are higher than 0.9 in all cases (world region and gender). The figures also demonstrate similarities and differences among the outcomes of the equations. For both gender groups, the highest values were presented for the 30–39 age group and decreased with age. For the 40–49 age group, the decline in strength capability was relatively mild, becoming more pronounced after the age of 50.

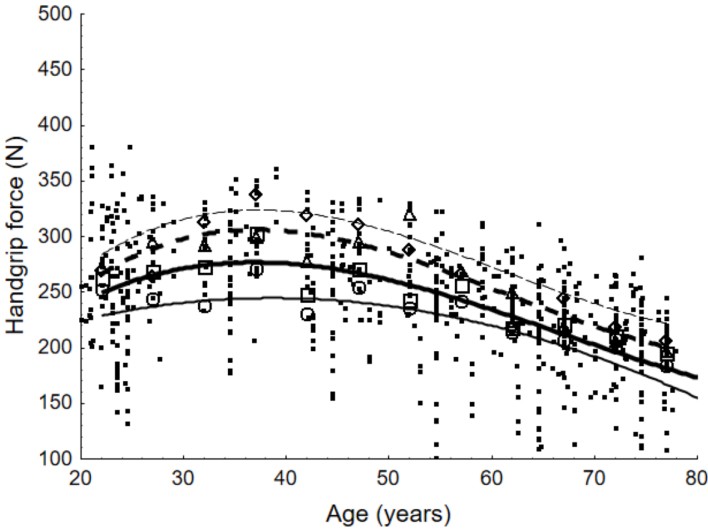

**Figure 5 Scatter points representing values of maximum handgrip force and weighted averages according to age group for female population and lines presenting force values determined with equations.** For the general population, weighted averages are denoted by a quadrat; for European countries, by a diamond; for the United States, by a triangle; and for Asian countries, by a circle. The equations that represent the maximum handgrip force as a function of age are illustrated by the lines in the figure. The equation for the general population is shown by the bold line. The countries of Europe, the United States, and Asia are represented by the next three lines, which are read from top to bottom.

Comparing maximum force capabilities among world regions for people aged 37 years old, the following become evident: European countries exhibit higher values than the general population by 17%, and the US by 10%. To the contrary, Asian countries present lower values by 12%. For the male population exclusively, the percentages relevant to the general population were 9%, 1% and 9% respectively.

Comparing the maximum handgrip force of 37-year-olds with that of 67-year-olds, the highest drop is observed in the US population and amounts to 24% for females and 23% for males. The lowest drop is reserved for Asian countries with 17% and 19% respectively.

Figure 7 presents the comparative results between the values of maximum handgrip force calculated for general male or female populations and the consolidated results obtained in the review conducted by *Bohannon et al. (2006)*. A five-year dispersion was included in this study. The Spearman's correlation coefficient equals 0.94 and 0.89 (with $p < 0.001$) for male and female populations respectively. It can be clearly noticed that there is strong compliance in the results. A difference of about 40N is observed for all age groups. The figure also illustrates differences in curves between male and female populations, testifying to the fact that decrease in force with age is steeper for males.

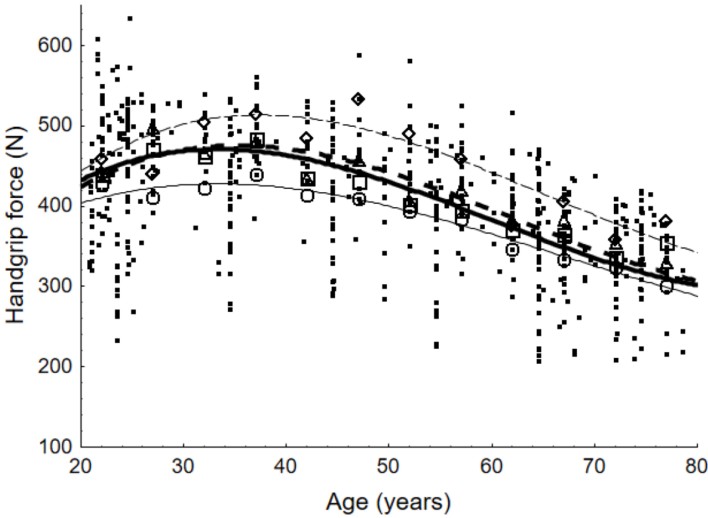

**Figure 6 Scatter points representing values of maximum handgrip force and weighted averages according to age group for male population and lines presenting force values determined with equations.** For the general population, weighted averages are denoted by a quadrat; for European countries, by a diamond; for the United States, by a triangle; and for Asian countries, by a circle. The equations that represent the maximum handgrip force as a function of age are illustrated by the lines in the figure. The equation for the general population is shown by the bold line. The countries of Europe, the United States, and Asia are represented by the next three lines, which are read from top to bottom.

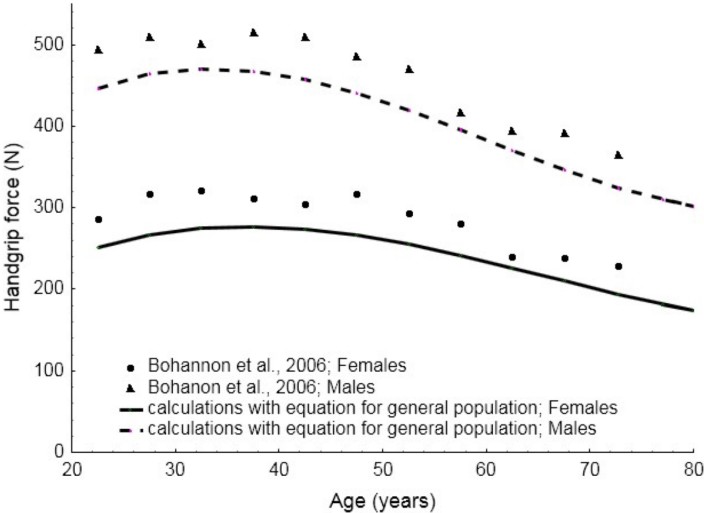

**Figure 7 Values of maximum handgrip force calculated with the equations developed in this study and the consolidated results presented by *Bohannon et al. (2006)*.**

## DISCUSSION

### Development of predictive equations

Utilising normative data, which presented handgrip force as a function of age, the purpose of this study was to develop predictive equations that were gender- and world region-specific. Quantitative mathematical equations of the third degree were developed as a result of the meta-analysis conducted based on a comprehensive literature review, which was supplemented by a quantitative analysis of the data. Equations correlate strongly with experimental data for the general population and world-specific populations for the EU, the US and Asia. Although experimental data covering more world regions has been published, there was sufficient data on which to venture equations only for those three areas. More evidence is needed from larger samples in order to develop such equations for more countries, representing other world regions.

For the development of the equations, data were integrated from studies on single and multiple age groups with measurements taken using the same procedure. Thus, there were several differences regarding populations and measurement conditions. The measurement of maximum handgrip force is influenced by specific experimental conditions, such as testing protocols and the equipment used, as the type of dynamometer alongside the diameter of the gripped sensor are significant factors (*Abaraogu et al., 2017*). Upper limb posture is also important for the measurement of force (*Roman-Liu, 2003*). Different values of maximum force, however, can be obtained in the same experimental conditions but for different populations of subjects, who naturally exhibit differences in their individual characteristics, such as forearm circumference and hand length (*Lopes et al., 2018*), body mass and height (*Hanten et al., 1999*; *Tveter et al., 2014*) or lifestyle (*Wiraguna & Setiati, 2018*; *Moratalla-Cecilia et al., 2016*).

Measurement-related and individual factors are significant and the effect of many of these on the exerted force is difficult to quantify. This means that even if a high number of such factors is included in the analysis, it would not be possible to take into account each and every one of them. Increasing the number of factors taken into account, however, reduces the amount of available data. Therefore, it seems that limiting the role of gender by taking a regional perspective into account, where there is sufficient data, is justified. A meta-analysis minimises errors resulting from discrepancies in the available data, as the larger the groups of data for analysis, the fewer the errors. The integration of a large amount of data makes up for differences among measurements that may surface in different studies, and yields fewer biased results than in the case of equations developed based on the results of a single study.

The results of the meta-analysis were compared with table of norms assigning handgrip force values to gender and age (measured in decades) presented in *Bohannon et al. (2006)* study. *Bohannon et al. (2006)* also presented a meta-analysis, which covered 12 studies that provided average values for general population for groups of males and females. The results of the present meta-analysis and the results of that by *Bohannon et al. (2006)* are very similar. Both exhibit very high correlation between the calculations of the equations and experimental data, which confirms the reliability of the former. In view of that, it can

be argued that the present comprehensive review and predictive equations are reliable since they are based on populations from many countries. Equations that can be representative of either general or specific populations can express quantitative differences in maximum handgrip force due to age, gender and world-region. In this respect, the results of the present study are innovative in the context of the existing body of literature.

## Comparison of handgrip force among age, gender and world region groups

The outcomes of the equations indicate that maximum strength capability occurs in the ages between 30 and 39 years old, which suggests that the highest muscle mass and strength are generally attained in one's forth decade. The calculations performed in this study show that a significant decrease in maximum force begins at about 50 years of age.

The developed equations confirm that handgrip force, as measured in females, is lower than in males, a fact probably due to differences in muscle mass between these populations (*Mathiowetz et al., 1985*; *Su et al., 1994*). The differences confirmed by the equations consist not only in a shift in values but are also demonstrated in the shape of the equation curves. This means that age influences strength capabilities differently in male and female populations. Maximum force values between males and females for the general population differed by about 40%. The difference was lowest for the 60–63 year olds and highest for the 18–25 year-olds.

The analysis also indicates very small differences between US and European populations and strong differences between those two and Asian populations. *Dodds et al. (2016)* indicated the same trends referring to the decrease in force capabilities with age and highlighted similarities between countries. They, however, stressed the importance of geographical region more from an economic rather than a national perspective.

The results of the calculations arising from the developed equations showed that the most pronounced differences were between genders, especially in the fourth decade of life. Differences due to age are high when comparing people in their fourth and fifth decades of life with those who are in their seventies. Age differences in maximum force capabilities are much lower when comparing people with two decades of difference between them. The lowest are differences due to world region, which, in the majority of cases, fall below 10%.

## The utility of the developed equations

The developed relationships can be used in work-related age management, which is increasingly important due to demographic processes that have resulted in an increased number of older employees in the labour force. The ageing of the population requires solutions that would maintain older people's capabilities and enable them to remain active in the labour market. Excessive strain is a result of the gap between employee capacity and work task requirements, and causes the development of symptoms of MSDs. In order to eliminate this gap, there are two ways to protect individuals. Age management can include musculoskeletal load and risk assessment and the resulting intervention on workstations or the transfer of older employees to less burdensome workstations, if necessary (*Roman-Liu, Tomasz Tokarski & Mazur-Rózycka, 2021*). Another option is to allow the equations to

determine the levels of force needed for rehabilitation training that will increase older employees' muscular endurance and strength.

Physical exertion during training programmes with a heavy load has beneficial effects on the human body (*Rooks et al., 2007*). Well-chosen exercise can improve physical fitness in the elderly. The systematic practice of training as a form of leisure activity contributes to functional changes in the body and is a means of counteracting and delaying the deterioration of the body (*Furrer et al., 2014*). Physical activity also reduces the development of MSDs, while regressive resistance strength training is an effective tool in reducing physical disability (*Liu & Latham, 2011*). In physical training, which can also be a part of rehabilitation procedures, it is important to select the quantity of the load carefully. The developed equations provide a tool for adjusting loads taking into account such individual factors as age and gender.

The assessment of musculoskeletal load and the risk of developing MSDs is an essential part of maintaining the good health of employees. As the risk of developing MSDs depends both on work tasks and on employees' physical capabilities, it is necessary to determine the abilities of older employees and to ensure that work tasks take particular note of their physical stamina, namely not only that of the general working population. Typically, methods for assessing the development of MSDs only takes into account biomechanical factors, bypassing individual (personal) capabilities (*Roman-Liu, 2013*). This means that for the elderly, whose force capabilities are lower, the actual load is higher than that estimated by current methods. In order to modify the latter to suit older employees, limit or reference force values must be scaled to reflect age-related changes in strength capabilities. The developed equations can serve this purpose.

## Strengths and limitations of this study

Limitations of the study might be linked to the fact that the results of measurements presented in the reviewed articles were aggregated, which could have been affected by many factors related to, for example research equipment, as well as to the studied groups. This could have influenced the results and the accuracy of the developed equations. Nevertheless, it seems that the aggregation of data from many individual studies, which is possible thanks to the meta-analysis, provides a clearer picture than a separate analysis of each individual study. It is common that a meta-analysis summarises research that does not necessarily apply to the same measurement methods or considers the same participants. Although the meta-analysis uses a number of simplifications, it has more power to detect important factors determining the outcome of an analysis. As mentioned above, it is to be expected that the larger the number of groups of data for analysis, the fewer the errors. This review analysed 143 publications, which seems to be a large enough number to minimise error. Additionally, in an effort to average handgrip force within age groups, each study was weighted by the number of samples. Therefore, the developed relationships can be considered as adequate indicators of the general trends in changes in maximum force values with age. These equations can be used to determine the predicted handgrip strength values for a person of a given age and gender, beyond a discrete set of data points from

norm tables. Equations can be programmed easily into the database software, which would make them easily applicable in clinical and research settings.

Another strength of this study is that it makes room to expand the developed relationship to other types of force. Some forces are exerted with the whole body, not only the upper limbs. Numerous studies, which have shown that handgrip strength correlated closely with measures of muscle strength from other muscle groups, including the lower limbs (*Samuel et al., 2012*), justify this attitude. Handgrip force measurements have frequently been used as a low-cost tool to predict overall total body strength surrogate measures (*Rantanen et al., 1998*; *Leong et al., 2015*). This measure also reflects an individual's general health and level of physical activity (*Leong et al., 2015*). Thus, changes in handgrip force can indicate changes in muscle groups in other body parts. As such, they can also describe employees' functional capabilities.

## CONCLUSION

The equations developed on the basis of maximum force values obtained from a comprehensive literature review proved that maximum handgrip force decreases with age. They indicated about 35% difference between the ages of 35 and 75 for both males and females. They also showed that the values of maximum handgrip force are about 40% higher for males than for females and that this difference changes with age. The decrease in force is slightly higher for the male population than for the female. The equations related to American, European and Asian populations proved that differences due to region are smaller than those due to age or gender. Age-related quantitative changes in the values of maximum force can be used in determining the appropriate level of force in rehabilitation training, which will increase older employees' muscular endurance, and also in creating standards or recommendations relevant to the design of their workstations.

### Funding

This article is based on the results of a research task carried out within the fifth stage of the National Programme "Improvement of safety and working conditions" supported within the scope of state services by the Minister responsible for labour (task no. 2.SP.22 entitled "Development of an interactive computer program supporting ergonomic intervention in the field of musculoskeletal loading of upper and lower limbs and the back"). The Central Institute for Labour Protection–National Research Institute is the Programme's main co-ordinator. There was no additional external funding received for this study. The funders had no role in study design, data collection and analysis, decision to publish, or preparation of the manuscript.

### Grant Disclosures

The following grant information was disclosed by the authors:
The fifth stage of the National Programme "Improvement of safety and working conditions" supported within the scope of state services by the Minister responsible for labour (task no.

2.SP.22 entitled "Development of an interactive computer program supporting ergonomic intervention in the field of musculoskeletal loading of upper and lower limbs and the back").

The Central Institute for Labour Protection–National Research Institute is the Programme's main co-ordinator.

## Competing Interests
The authors declare there are no competing interests.

## Author Contributions
- Danuta Roman-Liu conceived and designed the experiments, performed the experiments, analyzed the data, authored or reviewed drafts of the article, and approved the final draft.
- Joanna Kamińska conceived and designed the experiments, performed the experiments, analyzed the data, prepared figures and/or tables, and approved the final draft.
- Tomasz Macjej Tokarski conceived and designed the experiments, performed the experiments, analyzed the data, prepared figures and/or tables, and approved the final draft.

## Data Availability
The raw data are available in the Supplementary File.

## Supplemental Information
Supplemental information for this article can be found online at http://dx.doi.org/10.7717/peerj.17703#supplemental-information.

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
