# Peer review of "Population-specific equations of age-related maximum handgrip force: a comprehensive review"

_PeerJ, doi:10.7717/peerj.17703_

## Round 0.1 · original submission · Major Revisions

Dear Authors,

Please make revisions to your article according to the reviewers' comments or write a detailed rebuttal on a point-by-point basis.

**Language Note:** PeerJ staff have identified that the English language needs to be improved. When you prepare your next revision, please either (i) have a colleague who is proficient in English and familiar with the subject matter review your manuscript, or (ii) contact a professional editing service to review your manuscript. PeerJ can provide language editing services - you can contact us at copyediting@peerj.com for pricing (be sure to provide your manuscript number and title). – PeerJ Staff

Reviewer 1 ·

Basic reporting

The review is of broad and cross-disciplinary interest and within the scope of the journal. The review provides a comprehensive meta-analysis of studies performed in samples of participants of a large age range from different world regions. The introduction provides a clear rationale for the study.

Experimental design

No specific comment on study design.

Validity of the findings

Argumentative discussion meets the goals set out in the Introduction, limitations of the study are explained.

Additional comments

Minor corrections are needed:

Abstract:
Background, line 46 – do not use the abbreviation MSD in the Abstract, but the full word
Results:
Line 203 – whereas Tables 3 and 4 – add space between “3” and the word “and”
Line 204-216 – is it possible for the authors to refer (in brackets) to the respective studies mentioned throughout this paragraph?
Discussion:
Line 336 – please, note that the age range 30-39 years represents a person’s fourth (not third) decade
Line 351 – again, the authors probably refer to the third (not second) decade of life, take note also for the sentence in line 352!
Lines 378-379 – use the full word followed by the abbreviation (“musculoskeletal disorders (MSDs)”) only at the first mention of the word, after which it is enough to use only the abbreviation MSD
Line 400 – the review analyzed 146, not 116 publications. Please, correct the number.
Tables
Tables 1-4 – please, make a correction in the title of Tables 1-4, as they are the summaries of articles (or studies), not of “reviews” included in the meta-analysis

Reviewer 2 ·

Basic reporting

Language wise the text is correct ( I am not a native speaker though)
There is a sufficial background and a comprehensive research of literature involved.
The review is in cross disciplinary interest zone (occupational medicine and sports medicine as well as physiotherapy and orthopedics
The only problem in my opisnion that it does not add with too much novelty, there was an umbrella review in 2021
Pinar Soysal, Christopher Hurst, Jacopo Demurtas, Joseph Firth, Reuben Howden, Lin Yang, Mark A. Tully, Ai Koyanagi, Petre Cristian Ilie, Guillermo F. López-Sánchez, Lukas Schwingshackl, Nicola Veronese, Lee Smith,
Handgrip strength and health outcomes: Umbrella review of systematic reviews with meta-analyses of observational studies,
Journal of Sport and Health Science,
Volume 10, Issue 3,
2021,
Pages 290-295,
ISSN 2095-2546,

Experimental design

Design is logical and sources are cited exept for the above mentioned review

Validity of the findings

As this is a review , it meets needed goals and answeres some questions. I am not sure if there is an actual gap in that knowledge.

·

Basic reporting

• The manuscript is written in unambiguous, professional English, which is used throughout.

• The article includes sufficient introduction and background to demonstrate how the work fits into the broader field of knowledge. Relevant prior literature has been appropriately referenced.
• The article's structure is acceptable, with ‘standard sections.’ The Figures are relevant to the article's content and appropriately described and labeled. However, I have suggestions regarding the number of figures.
• The meta analysis is on broad, cross-disciplinary interest and within the journal's scope.
• The field has not been reviewed recently.
• In the introduction, the authors begin with the burden of musculoskeletal disorders (MSDs) and then attempt to explain how handgrip force has an indicator role. I would suggest the authors begin with hand grip power (HGF) and what are the usage of HGF. Then, introduce the factors that can influence the HGF and justify choosing the age and population (please try to incorporate ethnicity). Doing so would nicely create the stage to bring in the research question and objectives of the study to make it clear to the audience,

Experimental design

• The content is within the Aims and Scope of the journal.
• The authors have performed a rigorous investigation to a high technical & ethical standard.
• "Methods should be described with sufficient information to be reproducible by another investigator." This is my main focus. The authors have not described whether they followed any checklist (e.g., PRISMA), website, or software.
• In the question for bias, the authors have not mentioned which test of bias they adopted to screen the research in question. According to Page et. al (Page MJ, McKenzie JE, Bossuyt PM, Boutron I, Hoffmann TC, Mulrow CD, et al. The PRISMA 2020 statement: an updated guideline for reporting systematic reviews. BMJ 2021;372:n71. doi: 10.1136/bmj.n71), "Specify the methods used to assess risk of bias in the included studies, including details of the tool(s) used, how many reviewers assessed each study and whether they worked independently, and if applicable, details of automation tools used in the process." is mandatory.
• Used sources are adequately cited, quoted, or paraphrased.
• The review has been organized logically into coherent paragraphs/subsections.

Validity of the findings

There is a well-developed and supported argument that meets the goals set out in the Introduction.
The conclusion identifies unresolved questions/gaps / future directions.

Additional comments

The authors should use a forest plot to demonstrate the articles used in the meta-analysis.
The authors should produce grouped bar charts to show the comparison between males and females.

---

## Round 0.2 · accepted · Accept

Dear Authors,
Your paper is now acceptable for publication.

Reviewer 1 ·

Basic reporting

No further comment.

Experimental design

No further comment.

Validity of the findings

No further comment.